# Pick Your Poison: Molecular Evolution of Venom Proteins in Asilidae (Insecta: Diptera)

**DOI:** 10.3390/toxins12120738

**Published:** 2020-11-24

**Authors:** Chris M. Cohen, T. Jeffrey Cole, Michael S. Brewer

**Affiliations:** Howell Science Complex, East Carolina University, 1000 E 5th St., Greenville, NC 27858, USA; coleti16@students.ecu.edu

**Keywords:** Asilidae, transcriptome, positive selection

## Abstract

Robber flies are an understudied family of venomous, predatory Diptera. With the recent characterization of venom from three asilid species, it is possible, for the first time, to study the molecular evolution of venom genes in this unique lineage. To accomplish this, a novel whole-body transcriptome of *Eudioctria media* was combined with 10 other publicly available asiloid thoracic or salivary gland transcriptomes to identify putative venom gene families and assess evidence of pervasive positive selection. A total of 348 gene families of sufficient size were analyzed, and 33 of these were predicted to contain venom genes. We recovered 151 families containing homologs to previously described venom proteins, and 40 of these were uniquely gained in Asilidae. Our gene family clustering suggests that many asilidin venom gene families are not natural groupings, as delimited by previous authors, but instead form multiple discrete gene families. Additionally, robber fly venoms have relatively few sites under positive selection, consistent with the hypothesis that the venoms of older lineages are dominated by negative selection acting to maintain toxic function.

## 1. Introduction

Venoms are typically a composition of various neurotoxins, enzymes, ions, and small organic molecules [1,2,3,4]. Venom proteins generally originate via gene duplication of non-toxic proteins that are then selectively expressed in a venom gland (recruitment) and undergo subsequent neofunctionalization [3,5]. However, alternative processes such as single gene co-option or de novo protein evolution are also believed to be significant drivers of venom evolution in some taxa [6]. Venom proteins are often recruited from secretory proteins involved in rapid physiological processes and from those that have stable tertiary structures due to multiple disulfide bonds [7]. Among these polypeptide components, many neurotoxins have abundant inhibitory cystine knots (ICKs) while venom enzymes are larger and, owing to a different role in the venom cocktail, typically lack this motif [7].

Venom has independently evolved in at least four lineages of true flies (Insecta: Diptera) [8]. The family Asilidae (also known as robber flies or assassin flies) is unique among these in that the adults, rather than the larvae, are venomous predators. Their closest relatives, Apioceridae and Mydidae, as adults, either feed on nectar and other liquids or do not feed at all [9,10]. Adult assassin flies have a venom delivery apparatus rather unlike that found in most other venomous arthropods—the proboscis consists of the labium, which forms a tube through which the hypopharynx slides, and elements of the labrum and maxillae, which support the action of the hypopharynx [11,12]; venom is produced in a pair of thoracic glands (also called salivary glands) connected via a fused duct to the hypopharynx, and it is this structure that pierces prey and injects the saliva [12,13,14,15].

Robber flies are capable of incapacitating large and dangerous prey quite rapidly, depending on the site of venom injection [11,12,16,17], and they can deliver painful bites to humans [18,19,20,21]. Recent studies have described over 300 venom proteins and have organized novel venoms into fifteen ‘Asilidin’ protein families, using the taxa *Eutolmus rufibarbis*, *Machimus arthriticus*, *Dolopus genitalis*, and *Dasypogon diadema* [15,22,23]. Some of these venoms are species-specific, consistent with observations of variable venom toxicity between species demonstrated by previous authors [24,25,26,27,28,29]. Toxicity assays of crude venom or isolated components have shown paralytic and other neurotoxic effects in insect and mammalian subjects [15,22,24,30]. Multiple studies also indicate that disulfide-rich peptide scaffolds (e.g., ICKs), such as those found in spiders and scorpions, have been convergently recruited into the robber fly venom arsenal [15,30].

Sunagar and Moran developed a “two-speed” model of venom evolution in which the venom genes of young lineages (<60 myo) are often under strong positive selection as niche space is explored, while in older lineages (>380 myo), venom genes are typically dominated by purifying selection in order to preserve toxic function [31]. With the recent description of putative venom genes in Asilidae by Drukewitz et al. and Walker et al. (hereafter referred to as DEA and WEA, respectively [15,22]), it is now possible to examine the molecular evolution of venom in these flies, particularly the “two-speed” model of Sunagar and Moran. The age of the family Asilidae is estimated to be between 133–158 myo [32,33], similar to toxicoferan lizards at 166 myo [34]. Like the lizards analyzed in Sunagar and Moran’s study, we expect robber flies to show little evidence of positive selection in their venom genes. To test this hypothesis, we combine a novel whole-body transcriptome of *Eudioctria media* with 10 publicly available asiloid transcriptomes and provide them as input for the recently developed programs Families Under Selection in Transcriptomes (FUSTr) [35] and toxify [36] to identify putative venom gene families and determine the extent of pervasive positive selection.

## 2. Results

### 2.1. Quality, Assembly, and Completeness of Transcriptome

Illumina RNA sequencing generated 67.7 million raw reads for the *Eudioctria media* whole-body transcriptome. After preprocessing with trim galore (v0.3.7), 66.7 million reads (98%) remained. The *Eudioctria media* transcriptome was assembled into 103,352 contigs. BUSCO (v3.0.2) reported that the *E. media* transcriptome had 86% complete single-copy BUSCOs, 5.1% fragmented BUSCOs, and 8.2% missing BUSCOs. The statistics for this and the other transcriptomes are summarized in Table 1.

### 2.2. Site-Specific Signatures of Selection in Asilidae Venom Gene Families

The longest isoforms of 107,641 complete coding sequences were provided as input for FUSTr (v1.0). This identified 60,727 gene families, of which 348 contained ≥ 15 sequences (the minimum number allowing enough power for subsequent analyses). Of those 348 families, 77 contained at least one amino acid site under strong positive selection. FUSTr clustered the 308 venom-annotated genes and their homologs into 151 families, about 30% of which are singletons (i.e., gene families consisting of only one sequence). Sequences annotated as one of the 14 described asilidins by DEA/WEA were clustered into 34 families. Of those with more than two sequences, only asilidin5, asilidin7, asilidin12, and asilidin13 were recovered as individual, monophyletic families. Asilidin1, asilidin2, and asilidin10 were split the most with 5, 10, and 4 separate families, respectively. Asilidin14 was not included in analyses because Transdecoder (v3.0.1) could not find a complete open reading frame.

Twenty-three DEA/WEA-annotated venom families were large enough to be analyzed for evidence of selection. Eight of these families (34%) were found to be under positive selection. These contain homologs to asilidin2 and asilidin11 (both described from *D. genitalis*), dehydrogenase, deaminase, several peptidases, and alpha amylase. Nine additional putative venom gene families identified by toxify were also found to be under positive selection (see below for more details). These putative venom families under positive selection are shown in Figure 1. Venom dehydrogenase (family 2127) and peptidase M13 (family 696) are disregarded because they did not contain signal peptides, a necessary prerequisite for secreted proteins. Only two toxify-predicted families are shown.

A total of 12,976 coding sequences of the 107,641 provided as input for FUSTr were also classified as being a venom by toxify (v0.1.78) with a probability ≥ 0.90. Of the 348 gene families (with ≥15 sequences) analyzed by FUSTr, toxify classified 33 as having at least one protein sequence with a venom probability ≥ 0.90, and a total of nine of those families contained at least one amino acid site under positive selection. The nonvenomous outgroups *Apiocera parkeri* and *Mydas clavatus* were represented in 10 and 8 families of the 33, respectively. Furthermore, toxify classified 43 of 311 annotated proteins (22 of 151 putative gene families) from *E. rufibarbis*, *M. arthriticus*, and *D. genitalis* as venom proteins with a probability greater than 90% (e.g., asilidin1; Figure 2).

### 2.3. Gene Family Gain/Loss in Insects

The phylogeny inferred by Species Tree Inference from All Genes (STAG) (v1.0) is fully consistent with the topology produced by Dikow et al. (2017). The family Asilidae uniquely gained 2509 gene families, while the sister lineage Apioceridae + Mydidae gained 27 (Figure 3). The subfamily Asilinae, from which most recent venomic studies have been conducted (e.g., [15,22]), gained 251 gene families. Forty-three venom-annotated (DEA/WEA) gene families were recovered as present in the ancestor of Asiloidea, while forty venom-annotated gene families were gained in Asilidae. The subfamily Asilinae gained four venom-annotated gene families. No evidence for whole-genome duplications was recovered in any lineages comprising the Asiloidea.

## 3. Discussion

In this study, we used FUSTr to generate putative gene families derived from thoracic and whole-body transcriptomes and to identify which of these families had evidence of positive selection. We also used toxify to assign venom probabilities to these genes, and this is the first study of putative venom genes from across the family Asilidae. Previous studies have focused on only one or two species (e.g., [15,22,23]), but in this study, we included representatives from five subfamilies (including Asilinae), though this is still only a small fraction of the 14 currently recognized. We also, for the first time, examined venom gene family diversity in a broader phylogenetic context by including two representatives of the nonvenomous sister lineage to robber flies, Apioceridae + Mydidae.

FUSTr split most asilidins (all except 5, 7, 12, and 13) into multiple families, suggesting that these may not be natural groups, as defined by DEA and WEA. Both authors relied primarily on BLAST homology to delimit their respective asilidin families. However, it should be noted that the clustering algorithms implemented by FUSTr (i.e., SiLiX v1.2.11) have strict requirements for family assignments to reduce domain chaining, so families tend to have fewer sequences and tend to be more numerous. While we did not include data from Drukewitz et al. [23] (see methods), we noticed that those authors named new asilidin families (11–15), although asilidins 11–14 had already been described by Walker et al. [22]. This confused nomenclatural situation, in which the same asilidin family names are being used to refer to different gene families, will need to be addressed in the future.

This study shows that toxify can be a useful tool for identifying putative venom genes in understudied taxa for which existing genomic and/or proteomic resources are lacking. However, it should be considered as a complement to traditional venomics methods (i.e., identify candidate genes for further study), not a replacement. In Asiloidea, toxify appears to have both a high false negative and a high false positive rate. For example, only 14% of putative asilid venoms identified by previous authors [15,22] were predicted to be venoms by toxify. Conversely, of the gene families gained by the nonvenomous outgroups, *A. parkeri* and *M. clavatus*, 442 and 44 were predicted by toxify to contain venom sequences, respectively (Appendix A).

These results may be explained by the fact that the training data for toxify consisted largely of venoms from spiders, cone snails, and snakes, and this bias may limit its effectiveness outside of those groups. As more venoms from a wider array of animals become well-characterized, toxify will be retrained and its performance will likely improve.

Many putative venom gene families were already present in the ancestor of Asiloidea (41.5%), but a similar number were uniquely gained in Asilidae (39.6%). In contrast, many fewer (1.0–7.9%) venom gene families were gained in the various lineages of Asilidae studied here. This suggests that robber flies may use a suite of venoms that are fairly evolutionarily conserved across the family, supplemented by a small number of toxins unique to particular lineages. Drukewitz et al. found that 90% of asilid venom gene families were gained prior to the evolution of Asilidae [23], as opposed to our 41.5%. This difference may be due to our comparatively limited outgroup sampling, which may have obfuscated the true ancestry of some gene families, our expanded ingroup sampling, which may have recovered gene families not detected by Drukewitz et al., or inherent differences in clustering methods (Orthofinder version unreported vs. SiLiX v1.2.11), which can affect the number of gene families recovered.

The family Asilidae is roughly equivalent in age to the toxicoferan lizards included in the study of Sunagar and Moran, which had two out of six toxin families (33%) with sites under significant positive selection [31]. For comparison, eight of twenty-three (34%) of DEA/WES-annotated robber fly venom gene families showed evidence of positive selection. This indicates that venom gene sequence evolution in asilids, like that of other ancient venomous lineages, is dominated by purifying rather than positive selection.

Future venom studies in Asilidae should focus on transcriptomic and proteomic sequencing from phylogenetically disparate species, as well as toxicity assays of individual venom proteins. In addition, whole-genome sequencing of a phylogenetically diverse array of robber flies and their relatives will be necessary to properly explore the genomic processes involved [6,23]. Only with all these resources available will we be able to gain a truly comprehensive understanding of asilid venom diversity and its evolution.

## 4. Materials and Methods

All analyses use default parameters unless otherwise noted, and the coding sequences (CDS) generated for this study are available via DOI: 10.6084/m9.figshare.13130171.

### 4.1. Taxon Sampling

Eight high-quality thoracic asiloid transcriptomes and two asilid venom gland transcriptomes were downloaded from the NCBI Short Read Archive (see Table 1 for accession numbers). To this, we added an additional whole-body transcriptome from *Eudioctria media* (Asilidae: Dioctriinae) (NCBI SAMN13178732). This species was chosen because it represents a distinct lineage that is only distantly related to previously studied taxa [37]. Summary statistics for the retrieved transcriptomes can be found in Table 1. Transcriptomic data for *Dolopus genitalis* [22] were not available. Instead, 123 putative venom proteins sequences from this species were included in analyses of site-specific signatures of selection. Two other available robber fly transcriptomes representing important lineages (*Nicocles dives* and *Lasiopogon cinctus*) were not included because of low read counts and low BUSCO complete scores [33]. Venom gland transcriptomes of *Dasypogon diadema* [23] were not included because a close relative (*Diogmites neoternatus*) was already included.

### 4.2. RNA Isolation, Sequencing, and Processing

Whole body RNA was extracted from one adult male *E. media* using TRIzol (Life Technologies, Carlsbad, CA, USA) followed by purification using the Qiagen RNeasy kit (Qiagen, Valencia, CA, USA), following the manufacturer’s protocol. Whole-body RNA was used so that the resulting transcriptomic data would be useful for broader studies in asilid genomics. The RNA sample was sent to the Genomic Services Lab at HudsonAlpha (Huntsville, AL, USA) for library preparation and sequencing (100 bp; paired-end) on an Illumina HiSeq 2500.

Raw FASTQ files from all transcriptomes were provided as input for trim galore v0.3.7 [38] to trim low-quality reads and remove adapters. Trimmed reads were then assembled in Trinity v2.0.6 [39]. The program BUSCO v1.1b1 [40] was used to assess the transcriptome completeness of the assembly with the included Arthropoda dataset. Protein coding sequences were then extracted from the assembled transcripts using Transdecoder v3.0.1 [41].

### 4.3. Site-Specific Signatures of Selection in Asilidae Venom Gene Families

In order to characterize the molecular evolution of the robber fly venom genes described by DEA/WEA, transcriptome assemblies from nine Asilidae and two outgroups within Asiloidea (*Mydas clavatus* and *Apiocera parkeri*), as well as 123 putative venoms from WEA, were provided as input for FUSTr v1.0 [35] to detect gene families undergoing pervasive positive selection. FUSTr accomplished this by performing tests on codon alignments with the reconstructed phylogenies of gene families. Sequences were clustered into gene families using SiLiX (v1.2.11) [42]. To maintain minimum statistical power, only gene families with 15 or more sequences were analyzed for signatures of selection.

### 4.4. Venom Protein Family Assignment

Because nearly all of the described venom protein sequences from DEA/WEA were included in this analysis, we were able to classify families as probable “venoms” if they contained homologs to the previously described asilidin venom proteins. Additionally, we described novel putative venom gene families with toxify v0.1.78 [36], which uses a deep learning approach to infer whether a given protein sequence is a toxin. Putative venom proteins were only further characterized if the presence of a signal peptide was detected using SIGNALP v5.0 [43].

### 4.5. Gene Family Loss/Gain

In order to quantify the loss and gain of gene families in Asilidae in a phylogenetic framework, the gene family trees from FUSTr were provided as input to STAG v1.0 [44]. STAG takes gene trees from any multi-copy gene family that has all sequences from all species present and estimates divergence between each species pair from the closest estimated orthologous gene pairs. Through a consensus approach, STAG is then able to infer the species tree in a manner that has been demonstrated to be more accurate than multi-species coalescent-based approaches or concatenation. STAG is ideal for systems that have a history of duplication events that render one-to-one ortholog-only approaches unfeasible. To test whether asilid lineages have undergone substantial gains in gene families relative to other insects, we reconstructed ancestral gain and loss events for each FUSTr family for all branches in the inferred insect phylogeny using DOLLOP v3.69 from the PHYLIP package, [45] and supplementary Python scripts implemented through FUSTr (https://github.com.tijeco/FUSTr).

To investigate the origins of novel gene families and expansion of paralogs, thepackage DupPipe (EvoPipes.net) was employed on the authors’ server and results were delivered 14 December 2018. The dataset of Li et al. [46] was combined with the *E. media* transcriptome and the CDS sequences from the *Proctacanthus coquilletti* genome [33].

## Figures and Tables

**Figure 1 toxins-12-00738-f001:**
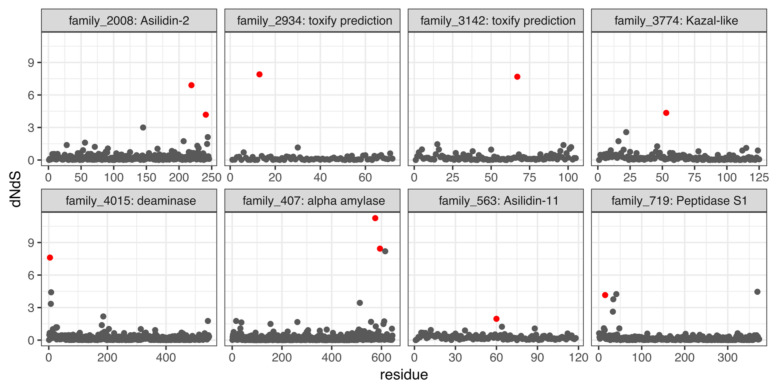
Site-specific synonymous to non-synonymous nucleotide substitution (dN/dS) ratios per amino acid position for eight putative venom gene families identified via Drukewitz et al. and Walker et al. (DEA/WEA) and toxify. Significant sites under positive selection are marked in red.

**Figure 2 toxins-12-00738-f002:**
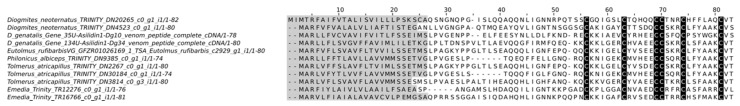
Multiple sequence alignment (MSA) of FUSTr family 2511, putative asilidin1. Signal sequences are highlighted in grey. Cysteine residues are highlighted in black.

**Figure 3 toxins-12-00738-f003:**
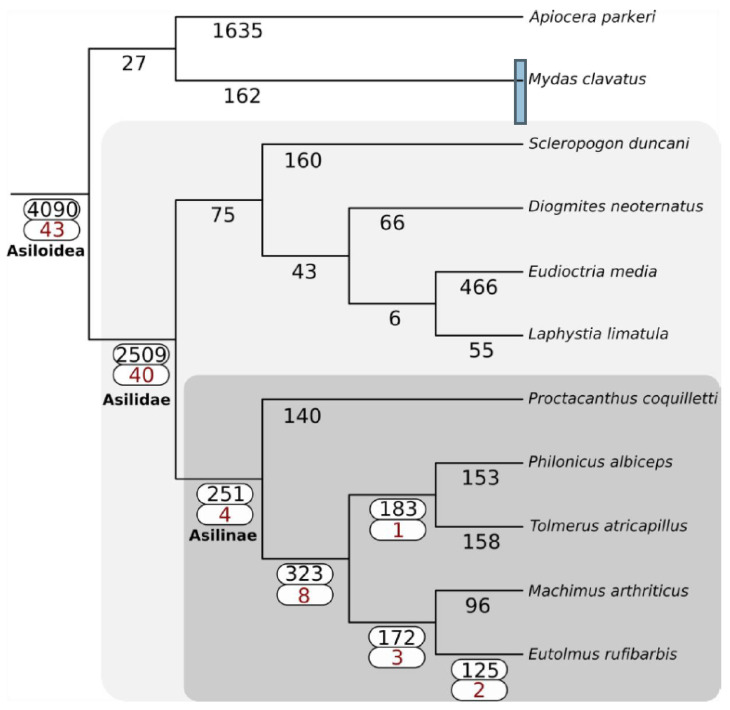
Gain of gene families across Asiloidea using SiLiX clustering and the principle of dollo parsimony as implemented in DOLLOP. The total number of gene families gained at each node is shown in black. For nodes with one or more DEA/WEA venom-annotated gene families gained (red), both values are highlighted with a white box.

**Table 1 toxins-12-00738-t001:** Processed reads, assembled transcripts, number of coding sequences (CDS), BUSCO complete percentage, and Sequence Read Archive (SRA) number for each taxon included in this study. Values for each transcriptome are derived from their respective publication.

Taxon	Family	Reads	Transcripts	CDS	BUSCO	SRA
*Apiocera parkeri*	Apioceridae	143,373,948	298,313	112,400	65%	SRX2213179
*Diogmites neoternatus*	Asilidae	120,499,106	43,199	18,145	80%	SRX2212172
*Eudioctria media*	Asilidae	133,342,485	103,000	37,075	86%	SRR10386632
*Eutolmus rufibarbis*	Asilidae	87,187,856	56,640	15,055	94.50%	SRX2501518
*Laphystia limatula*	Asilidae	60,777,554	30,019	20,511	25%	SRX2213169
*Machimus arthriticus*	Asilidae	83,421,201	69,849	16,916	90%	SRX2501520
*Mydas clavatus*	Mydidae	90,390,602	54,643	26,033	63%	SRX2212335
*Philonicus albiceps*	Asilidae	107,425,636	46,977	27,750	90%	SRX2222448
*Proctacanthus coquilletti*	Asilidae	21,978,654	56,925	36,770	79%	SRX2213473
*Scleropogon duncani*	Asilidae	111,276,014	50,672	25,148	69%	SRX2213475
*Tolmerus atricapillus*	Asilidae	108,444,670	43,915	23,324	90%	SRX2213152

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
