# Peer review of "Pick Your Poison: Molecular Evolution of Venom Proteins in Asilidae (Insecta: Diptera)"

_toxins, 2020, doi:10.3390/toxins12120738_

Round 1

Reviewer 1 Report

The quality of the manuscript is excellent.

However, I wonder if the essence of this paper is a subject that should be covered by Toxins. Perhaps a more appropriate journal in the field of computational science would be a better place for this paper. The author also stated that some of the results contradicted with those of previous studies and questioned the validity of the calculation conditions.

Finally, there is no doubt that the authors' work of Asilidae venom has made a great contribution to our field of venom research.

I am very much looking forward to seeing what venomous activity the proteins encoded by the transcripts they predicted to be venomous will show in the venom assay.

Author Response

Point 1: However, I wonder if the essence of this paper is a subject that should be covered by Toxins. Perhaps a more appropriate journal in the field of computational science would be a better place for this paper. The author also stated that some of the results contradicted with those of previous studies and questioned the validity of the calculation conditions.

Response 1: We believe that the subject matter of our paper, venom evolution is Asilidae, conforms well with the Aims of Toxins.

Reviewer 2 Report

The authors identified, using transcriptome analysis and bioinformatic tools, gene families that were predicted to contain venom genes in genetic material from Eudiocrita media (robber flies). They also suggest the possible molecular evolutionary mechanisms (positive and negative selection) of the genes related with venom’s proteins. Since the venom characterization of the asilid species was recently determined the better comprehension of the possible mechanisms by which these species acquired and/or conserved these genes is considerably relevant. In this sense, we would like to ask some questions an suggest the following modifications.

Results section

1 -Have the authors published the sequences obtained in transcriptome analysis in NCBI bank or another database? Please provide the SRA entries or similar accession number.

2- Please provide a table that summarizes the protein sequences that presents a high probability to be a venom protein, as described in line 100-101.

3- Line 66 – put the specie name in italic format

Discussion section

Write the discussion section as a continuous text, do not divide into subitems.

Methods section

  • Please provide the information described in line 179 and 180 since they are not accessible by the informed DOI number.     
  • Line 194 – correct TRIzolà – TRIzol
  • Line 195 – insert the information that the procedures followed the manufacturer’s instructions and inform eventually modifications of the RNA extraction protocols.
  • Line 195-196 – change “the extraction” to “RNA sample”

Author Response

Point 1: Results section: Have the authors published the sequences obtained in transcriptome analysis in NCBI bank or another database? Please provide the SRA entries or similar accession number.

Response 1: The novel transcriptome generated for this study (Eudioctria media) was submitted to the GenBank SRA (SAMN13178732; see Table 1). The cds sequences used for our analyses can be found at the provided figshare DOI and will be made available once the article is published.

Point 2: Results section: Please provide a table that summarizes the protein sequences that presents a high probability to be a venom protein, as described in line 100-101.

Response 2: These lines refer to 33 gene families that contain a total of 259 protein sequences, too many to be summarized in a table in the main manuscript. This information can be found in Table S1 by using filtering parameters in Excel (toxify_prediction -> greater than or equal to 0.9; n_fams -> greater than or equal to 15).

Point 3: Results section: Line 66 – put the specie name in italic format

Response 3: We have corrected the italicization of "Eudioctria media".

Point 4: Discussion section: Write the discussion section as a continuous text, do not divide into subitems.

Response 4: We have removed the subsections from the discussion.

Point 5: Methods section: Please provide the information described in line 179 and 180 since they are not accessible by the informed DOI number.

Response 5: The word "results" has been replaced with "the coding sequences (CDS) generated for this study" to clarify what is being provided. The figshare DOI will be made available once the article is published.

Point 6: Methods section: Line 194 – correct TRIzolà – TRIzol

Response 6: We have corrected the misspelling of TRIzol.

Point 7: Methods section: Line 195 – insert the information that the procedures followed the manufacturer’s instructions and inform eventually modifications of the RNA extraction protocols.

Response 7: We added " following the manufacturer's protocol" to the end of the sentence.

Point 8: Methods section: Line 195-196 – change “the extraction” to “RNA sample”

Response 8: We replaced "extraction" with "RNA sample".

Reviewer 3 Report

The manuscript titled “Pick Your Poison: Molecular Evolution of Venom Proteins in Asilidae (Insecta: Diptera)” describes the evolution of venom components in robber flies. This is an interesting study as it reveals an “intermediate” evolutionary rate that correlates with the evolutionary age of venom in this insect lineage. However, I did encounter some issues in this manuscript that I believe should be addressed before it can be further considered for publication.

Major comments:

  1. In lines 160-163 the authors mention that their results are highly contrasting to the results by Drukewitz et al. (Gigascience 8, 2019). Their explanations for this discrepancy are minimal and not very clear. More detailed explanations should be provided.
  2. The authors extracted RNA from whole bodies of robber flies rather than from venom-related organs. This choice might heavily mask venom genes and the authors at the very minimum explain why this was done. Is this a technical limitation? If yes, just say so. Explanations are important.

Minor comments:

  1. Line 12: Homologs to previously described toxins or venom proteins, but surely not venoms.
  2. Lines 13-14: The phrasing here is quite confusing. Please rephrase this sentence.
  3. Lines 30-31: The sentence regarding ICK motifs is not very clear. Please rephrase.
  4. Table 1: The first family name is not fitting into one line. Same is true for the third SRA accession number. Please fix this.
  5. Figure 2: It would be helpful to use colors to highlight some additional conserved positions.
  6. Lines 138-140: It is unclear why the authors did not include those data from Drukewitz et al. Please provide some explanation for this choice. Moreover, it is somewhat unclear what is the confusion in the naming the authors are referring to. Please try to explain more clearly.

Author Response

Point 1: In lines 160-163 the authors mention that their results are highly contrasting to the results by Drukewitz et al. (Gigascience 8, 2019). Their explanations for this discrepancy are minimal and not very clear. More detailed explanations should be provided.

Response 1: We have clarified this section and it now reads thusly, "Drukewitz et al. found that 90% of asilid venom gene families were gained prior to the evolution of Asilidae [23], as opposed to our 41.5%. This difference may be due to our comparatively limited outgroup sampling which may have obfuscated the true ancestry of some gene families, our expanded ingroup sampling which may have recovered gene families not detected by Druketwitz et al, or inherent differences in clustering methods (Orthofinder (version unreported) vs. SiLiX (v1.2.11)) which can affect the number of gene families recovered."

Point 2: The authors extracted RNA from whole bodies of robber flies rather than from venom-related organs. This choice might heavily mask venom genes and the authors at the very minimum explain why this was done. Is this a technical limitation? If yes, just say so. Explanations are important.

Response 2:  The following sentence was added " Whole body RNA was used so that the resulting transcriptomic data would be useful for broader studies in asilid genomics." Sequencing of a whole body transcriptome would affect relative expression levels of venom genes but not necessarily hinder recovery of putative venom genes, which would still be present in the RNA sample and subsequently sequenced. We did not study expression levels, and so we do not believe that the whole body transcriptome negatively affected our results.

Point 3: Line 12: Homologs to previously described toxins or venom proteins, but surely not venoms.

Response 3: We changed "venoms" to "venom" and added the word "proteins" after "venom" on line 12, so now this sentence reads, "We recovered 151 families containing homologs to previously described venom proteins, and 40 of these were uniquely gained in Asilidae."

Point 4: Lines 13-14: The phrasing here is quite confusing. Please rephrase this sentence.

Response 4: This sentence now reads, "Our gene family clustering suggests that many asilidin venom gene families are not natural groupings as delimited by previous authors, but instead form multiple discrete gene families."

Point 5: Lines 30-31: The sentence regarding ICK motifs is not very clear. Please rephrase.

Response 5: The sentence now reads " Among these polypeptide components, many neurotoxins have abundant inhibitory cysteine knots (ICKs) while venom enzymes are larger and, owing to a different role in the venom cocktail, typically lack this motif [7]."

Point 6: Table 1: The first family name is not fitting into one line. Same is true for the third SRA accession number. Please fix this.

Response 6: We have reformatted the column widths of Table 1 so that "Apioceridae" and "SAMN13178732" are each on one line.

Point 7: Figure 2: It would be helpful to use colors to highlight some additional conserved positions.

Response 7: Figure 2 is only concerned with highlighting cysteine residues, which directly affect ICK formation. Other conserved regions are not particularly relevant.

Point 8: Lines 138-140: It is unclear why the authors did not include those data from Drukewitz et al. Please provide some explanation for this choice.

Response 8: The reasoning for this exclusion is discussed in the methods, so "(see methods)" was added.

Point 9: Moreover, it is somewhat unclear what is the confusion in the naming the authors are referring to. Please try to explain more clearly.

Response 9: We added: ", in which the same asilidin family names are being used to refer to different gene families," to this section.

Round 2

Reviewer 3 Report

The authors have addressed my remarks in an adequate manner. I do not have further remarks.